# Insulin-Like Growth Factor-1 Influences Prostate Cancer Cell Growth and Invasion through an Integrin α3, α5, αV, and β1 Dependent Mechanism

**DOI:** 10.3390/cancers14020363

**Published:** 2022-01-12

**Authors:** Carolin Siech, Jochen Rutz, Sebastian Maxeiner, Timothy Grein, Marlon Sonnenburg, Igor Tsaur, Felix K.-H. Chun, Roman A. Blaheta

**Affiliations:** 1Department of Urology, Goethe-University, 60590 Frankfurt am Main, Germany; Carolin.Siech@kgu.de (C.S.); Jochen.Rutz@kgu.de (J.R.); sebastian.maxeiner@kgu.de (S.M.); timo.grein@hotmail.de (T.G.); marso97@t-online.de (M.S.); felix.chun@kgu.de (F.K.-H.C.); 2Department of Urology and Pediatric Urology, University Medicine Mainz, 55131 Mainz, Germany; Prof.Dr.med.Igor.Tsaur@unimedizin-mainz.de

**Keywords:** prostate cancer, IGF-1, Akt-mTOR pathway, integrins, growth, invasion

## Abstract

**Simple Summary:**

Insulin-like growth factor-1 (IGF-1) is a growth hormone and is implicated in prostate cancer progression. Most prostate cancers begin in an androgen-dependent state so that androgen deprivation therapy results in improved clinical outcome. However, some cancerous cells may survive androgen deprivation, growing into therapy-resistant, androgen-independent prostate cancer. The present study investigated the influence of IGF-1 on tumor growth and migration properties using androgen-dependent LNCaP and VCaP and androgen-independent PC3 and DU145 prostate cancer cells. Stimulation with IGF-1 activated growth in all cell lines. There were changes in transmembrane receptors (integrins) that bind cells to each other and changes in focal adhesion kinase that controls cell motility. Intracellular Akt/mTOR signaling, regulating cell division, was also activated. Thus, it seems that prostate cancer progression is controlled by a fine-tuned network between IGF-1-driven integrin-FAK signaling and the Akt-mTOR pathway. Concerted targeting of both pathways may, therefore, help prevent cancer dissemination.

**Abstract:**

Insulin-like growth factor-1 (IGF-1)-related signaling is associated with prostate cancer progression. Links were explored between IGF-1 and expression of integrin adhesion receptors to evaluate relevance for growth and migration. Androgen-resistant PC3 and DU145 and androgen-sensitive LNCaP and VCaP prostate cancer cells were stimulated with IGF-1 and tumor growth (all cell lines), adhesion and chemotaxis (PC3, DU145) were determined. Evaluation of Akt/mTOR-related proteins, focal adhesion kinase (FAK) and integrin α and β subtype expression followed. Akt knock-down was used to investigate its influence on integrin expression, while FAK blockade served to evaluate its influence on mTOR signaling. Integrin knock-down served to investigate its influence on tumor growth and chemotaxis. Stimulation with IGF-1 activated growth in PC3, DU145, and VCaP cells, and altered adhesion and chemotactic properties of DU145 and PC3 cells. This was associated with time-dependent alterations of the integrins α3, α5, αV, and β1, FAK phosphorylation and Akt/mTOR signaling. Integrin blockade or integrin knock-down in DU145 and PC3 cells altered tumor growth, adhesion, and chemotaxis. Akt knock-down (DU145 cells) cancelled the effect of IGF-1 on α3, α5, and αV integrins, whereas FAK blockade cancelled the effect of IGF-1 on mTOR signaling (DU145 cells). Prostate cancer growth and invasion are thus controlled by a fine-tuned network between IGF-1 driven integrin-FAK signaling and the Akt-mTOR pathway. Concerted targeting of integrin subtypes along with Akt-mTOR signaling could, therefore, open options to prevent progressive dissemination of prostate cancer.

## 1. Introduction

Prostate cancer is the second most common cancer in men worldwide, with 1.3 million new cases registered in 2018, and a leading cause of cancer death [1]. The transformation of normal epithelial cells into malignant tumor cells takes place in the following two fundamental steps: initiation of uncontrolled proliferation and activation of motile spreading of initially healthy, highly differentiated cells. Both processes speed up the mitotic cell cycle and, once the primary tumor has been established, force the motile machinery of the tumor cells to allow spreading into distant organs. A key role in prostate cancer growth activity is played by the insulin growth factor 1 receptor (IGF-1R) with subsequent phosphorylation of the downstream targets, Akt and mechanistic target of rapamycin (mTOR) [2,3]. A clinical trial has demonstrated a close association between circulating IGF-1 and prostate cancer development and progression [2].

Escape of single cells from the primary tumor and resettling as secondary tumors requires interaction with vascular endothelial cells and extracellular matrix substrates such as collagen or fibronectin. These actions are mediated by adhesion receptors of the integrin family [4,5], namely heterodimer cell surface molecules, consisting of α- and β-subunits. Following cell–cell- or cell–matrix-binding, integrin-dependent downstream signaling is established via kinases and signaling adaptors, including focal adhesion kinase (FAK), a cytoplasmic non receptor tyrosine kinase [6,7]. A multivariable-adjusted Cox regression carried out on 199,698 prostate cancer patients has revealed a close relationship between integrin expression and tumor colonization of the bone microenvironment [8].

Prostate cancer growth and invasion is complex and tightly coordinated, controlled by intracellular signaling cascades that have controversially been specified. Caromile et al. observed signaling between β1 integrin and IGF-1R [9]. Akt has been shown to act on apoptosis in concert with integrin α5, although separate activity of either Akt or integrin α5 on apoptosis has been demonstrated as well [10]. A humanized monoclonal antibody against integrin αV decreased phosphorylation of Akt in prostate cancer cells [11]. Others, however, have not seen an Akt-αV integrin-interaction in prostate cancer cells [12].

Using human androgen-independent prostate cancer cell lines (DU-145, PC3) as well as androgen-sensitive LNCaP and VCaP cells, the present study was designed to evaluate how IGF-1 is involved in integrin driven regulation of tumor cell adhesion and migration and, conversely, whether Akt/mTOR signaling activated by IGF-1 is involved in integrin driven regulation of tumor growth.

## 2. Materials and Methods

### 2.1. Cell Lines

The human prostate tumor cell lines DU145, PC3, and LNCaP were obtained from DSMZ (Braunschweig, Germany). VCaP cell lines were from the Department of Urology and Pediatric Urology, Saarland University, Homburg/Saar, Germany. DU145, PC3, and LNCaP tumor cell lines were grown in RPMI 1640 medium (Gibco/Invitrogen, Karlsruhe, Germany) supplemented with either 2 or 10% fetal bovine serum (FBS), 2% HEPES (2-[4-(2-hydroxyethyl)piperazin-1-yl]ethanesulfonic acid) buffer (1 M, pH 7.4), 1% GlutaMAX, and 1% penicillin/streptomycin (all: Gibco/Invitrogen) at 37 °C in a humidified incubator with 5% CO_2_. VCaP were grown in DMEM medium, supplemented with FBS, 1% penicillin/streptomycin, 2% GlutaMAX and 1% sodium pyruvate (all: Gibco/Invitrogen).

### 2.2. IGF-1R Detection and IGF-1 Stimulation

Surface expression of the insulin-like growth factor 1 receptor (IGF-1R) was evaluated by flow cytometric analysis using the monoclonal antibody phycoerythrin (PE)-conjugated anti Human CD221 (IGF-1R; clone1H7) or the Alexa Fluor 647-labelled phospho-IGFIR (pIGF-1R, IgG1, pY1131, clone K74-218; both BD Pharmingen). Mouse IgG1-PE (κ Isotype, Clone MOPC-31C) and IgG1-Alexa Fluor 647 (κ Isotype, Clone MOPC-31C; all: BD Bioscience) served as the control isotypes. Receptor surface expression was measured using a FACSCanto (BD Biosciences, Heidelberg, Germany; FL-2H or FL-4H (log) channel histogram analysis; 1 × 10^4^ cells per scan) and expressed as mean fluorescence units (MFU). For stimulation studies, tumor cells were incubated with LONG^®^R3IGF-1 (IGF-1; Sigma Aldrich, München, Germany) at a concentration of 100 ng/mL for different time periods as indicated.

### 2.3. Tumor Cell Growth

The 3-(4,5-dimethylthiazol-2-yl)-2,5-diphenyltetrazolium bromide (MTT) dye reduction assay (Roche Diagnostics, Penzberg, Germany) was used to evaluate cell growth. For each cell line, 5000 cells were pipetted in triplicate to 96-well plates with medium enriched with soluble IGF-1. Control cells received cell culture medium without IGF-1 and remained untreated. After 24, 48, and 72 h, 10 µL MTT (0.5 mg/mL) was added for an additional 4 h. Cells were then lysed overnight in solubilization buffer (10% SDS in 0.01 M HCl) at 37 °C, 5% CO_2_. Absorbance at 550 nm was assessed with a microplate enzyme-linked immunosorbent assay (ELISA) reader. To correlate absorbance with cell number, a defined number of cells ranging from 2500 to 160,000 cells/well was added to the microtiter plates (in triplicate). After subtracting background absorbance (cell culture medium alone), results were expressed as mean cell number.

### 2.4. Tumor Cell Adhesion and Chemotaxis

Tumor cells were incubated with 100 ng/mL IGF-1 for 4 or 24 h (versus unstimulated controls) and were then subjected to adhesion or chemotaxis assays. To analyze tumor cell adhesion, 6-well plates (Falcon Primaria; Corning, Wiesbaden, Germany) were coated with collagen G (matrix mixture consisting of 90% collagen type I and 10% collagen type III; Biochrom, Berlin, Germany) or fibronectin (derived from human plasma; BD Biosciences, Heidelberg, Germany) overnight. Plastic wells were used as background controls. Nonspecific cell adhesion was prevented by washing the plates with PBS and incubating with 1% bovine serum albumin (BSA) in PBS for 60 min. An amount of 2.5 × 10^5^ tumor cells/mL (0.5 mL/well) were then added to each well for 1 h at 37 °C. Following 1 h incubation, non-adherent tumor cells were removed by repeated washing with PBS with Ca^2+^ and Mg^2+^. The remaining adherent cells were fixed using 1% glutaraldehyde (Sigma, München, Germany). The mean cellular adhesion rate was calculated microscopically by counting five different fields (each 0.25 mm^2^) with a raster ocular at 200-fold magnification.

Chemotactic movement of tumor cells was examined by a Boyden double chamber system with 8 µm pore filters (6-well chamber system; Greiner Bio-One, Frickenhausen, Germany). Tumor cells pre-exposed to 100 ng/mL IGF-1 for 4 or 24 h (controls did not receive IGF-1) were detached from the culture flasks and placed in the upper chamber with serum-free medium (2.5 × 10^5^ cells/mL; 2 mL cell suspension/chamber). The lower chamber contained 10% FCS as the chemoattractant. After 24 h incubation time, migrated cells under the membrane were fixed with 1% glutaraldehyde and stained with haematoxylin (Sigma). Non-migrated cells were cleaned from the upper surface of the membrane with cotton-wool tips. The migrated cells were counted microscopically in five different observation fields under a microscope at 200-fold magnification (5 × 0.25 mm^2^).

### 2.5. Scratch Wound Assay

The scratch wound assay was used to examine the horizontal migration of the cancer cells in the presence of IGF-1 (versus controls). Tumor cells were incubated with 100 ng/mL IGF-1 at 37 °C, 5% CO_2_ for 4 or 24 h and then seeded onto 96-well ImageLock plates (Sartorius, Goettingen, Germany) previously coated with 400 µg/mL collagen at 4 °C for 48 h (100 µL cell suspension, 5 × 10^5^ cells/mL). At 24 h after plating out the cells, a defined scratch of about 700 µm was made with an IncuCyte^®^ WoundMaker (Sartorius). Detached cells were removed by washing with PBS with Ca^2+^ and Mg^2+^. Cell culture medium was then renewed with 200 µL medium with 2 or 10% FCS with 100 ng/mL IGF-1. Controls received cell culture medium without IGF-1. Plates were incubated in Incucyte^®^ Zoom (Sartorius) at 37 °C, 5% CO_2_ and photographed every 2 h for 48 h. Each experiment was conducted in triplicate. Relative wound density was calculated by WimScratch software (Onimagin Technologies SCA, Córdoba, Spain).

### 2.6. Integrin Surface Expression

Cancer cells stimulated with IGF-1 for 2, 4, or 24 h were detached from the culture flasks using Accutase® (PAA Laboratories GmbH, Pasching, Austria) and washed with blocking solution (PBS, 0.5% BSA). Subsequently, they were incubated for 1 h at 4 °C with phycoerythrin (PE) conjugated monoclonal antibodies (each 20 µL) directed against the following integrin subtypes: anti-α1 (IgG1; clone SR84, dilution 1:1000), anti-α2 (IgG2a; clone 12F1-H6, dilution 1:250), anti-α3 (IgG1; clone C3II.1, dilution 1:1000), anti-α5 (IgG1; clone IIA1, dilution 1:5000), anti-α6 (IgG2a; clone GoH3, dilution 1:200), anti-β1 (IgG1; clone MAR4, dilution 1:2500), anti-β3 (IgG1; clone VI-PL2, dilution 1:2500), anti-β4 (IgG2a; clone 439-9B, dilution 1:250; all: BD Biosciences), or anti-αV (IgG1; clone 13C2, Southern Biotech, Birmingham, AL, USA). The integrin expression of the tumor cells was then measured and analyzed using a FACSCalibur (BD Biosciences; FL2-H (log) channel histogram analysis; 1 × 10^4^ cells per scan) and expressed as mean fluorescence units. Mouse IgG1-PE (MOPC-21), mouse IgG2a-PE (G155-178) or rat IgG2b-PE (R35-38; all: BD Biosciences) were used as isotype controls.

### 2.7. Western Blot Analysis

To explore integrin protein expression as well as proteins involved in the Akt-mTOR-pathway under IGF-1 stimulation, tumor cell lysates were applied to a 7–12% polyacrylamide gel, electrophoresed for 90 min at 100 V and transferred to nitrocellulose membranes for 60 min at 100 V. After blocking with non-fat dry milk for 60 min, membranes were incubated overnight with the following unconjugated monoclonal antibodies, directed against integrin proteins and integrin related signaling, as follows: anti-α3 (polyclonal, dilution 1:500), anti-α5 (clone 1/CD49e, dilution 1:5000), anti-β1 (clone 18/CD29, dilution 1:2500; all from BD Biosciences), and anti-αV (Clone 21/CD51, dilution 1:250; Southern Biotech). Anti-FAK (clone 77, dilution 1:1000), anti-pFAK (pY397; clone 18, dilution 1:1000), and anti-ILK (clone 3, dilution 1:1000) antibodies were also from BD Biosciences. The mTOR pathway was investigated through the following proteins: Anti-Rictor (clone D16H9, dilution 1:1000), anti-pRictor (Thr1135, clone D30A3, dilution 1:1000), anti-Raptor (clone 24C12, dilution 1:1000), anti-pRaptor (Ser792, dilution 1:1000; all from Cell Signaling, Cambridge, UK), PKBα/Akt (IgG1, clone 55), and pAkt (IgG1, pS472/pS473, clone 104A282; both from BD Biosciences).

Horseradish peroxidase (HRP)-conjugated goat-anti-mouse or goat-anti-rat-IgG (Cell Signaling Technology, Cambridge, UK; dilution 1:3000) served as the secondary antibodies. To visualize proteins, membranes were incubated with enhanced chemiluminescence (ECL) detection reagent (ECLTM, Amersham/GE Healthcare, München, Germany). After incubation they were analyzed by the Fusion FX7 system (Peqlab, Erlangen, Germany). β-actin (1:1000; Sigma, Taufenkirchen, Germany) served as internal control. Pixel density analysis of the protein bands was performed with Gimp 2.8.20 software, www.gimp.org) before calculating the ratio of protein intensity/β-actin intensity.

### 2.8. Blocking Studies and siRNA Knock Down

Tumor cells were incubated for 60 min with 10 μg/mL function-blocking anti-integrin α3, α5, αV or anti-integrin β1 mouse mAb (all from Merck Millipore, Darmstadt, Germany) and then subjected to the MTT-assay and the chemotaxis assay. In addition, DU145 cells were treated with 10 µM of the FAK-inhibitor defactinib (Biozol, Eching, Germany) for 24 and 72 h. The mTOR related signaling proteins, Rictor and Raptor, both total and phosphorylated, were then evaluated with Western blotting. Transfection with small interfering RNA (siRNA) was carried out directed against integrin α3 (gene ID: 3675; target Sequence: 5′-CACCATCAACATGGAGAACAA-3′, α5 (gene ID: 3678, GeneGlobe ID-SI02654841, NM_002205) αV (gene ID: 3685; target sequence: 5′-TAGCATGATGTTACAGGAATA-3′) or anti-integrin β1 (gene ID: 891, target sequence: 5′-AATGTAGTCATGGTAAATCAA-3′) as well. Further studies were completed with siRNA directed against FAK (gene ID: 2185; target sequence: AAGCTGATCGGCATCATTGAA) or Akt (gene ID: 207; target sequence: AATCACACCACCTGACCAAGA; all: Qiagen, Hilden, Germany). An amount of 3 × 10^5^ cells were pre-incubated for 24 h with a transfection solution of siRNA and transfection reagent (HiPerFect Transfection Reagent; Qiagen) at a ratio of 1:6. Cells treated with cell culture medium alone and cells treated with 5 nM control siRNA (AllStars Negative Control siRNA; Qiagen) served as controls. Protein expression and tumor cell growth and chemotaxis were then analyzed as described above.

### 2.9. Statistics

The mean +/− SD was calculated. Graphs were prepared using SigmaPlot 11 (SYSTAT Software, San Jose, CA, USA). To exclude coincidence, all experiments were repeated three to five times. Statistical significance was evaluated with the “Student’s *t*-Test”. *p* < 0.05 was considered significant.

## 3. Results

### 3.1. IGF-1 Activates Tumor Cell Growth

Before exposing the tumor cells to IGF-1, expression of IGF-1R (DU145) and pIGF-1R (DU145, PC3) was verified by Facs analysis. Figure 1 demonstrates that both IGF-1R and pIGF-1R are present in DU145 cells, whether the cells were incubated in FBS-free or FBS-containing medium. pIGF-1R was also detected in PC3 cells. IGF-1R expression (total and phosphorylated) was verified in the androgen-sensitive LNCaP and VCaP cell lines as well (Figure 1).

Since FBS may influence tumor cell growth and mask IGF-1 specific effects, tumor cell growth activity was evaluated in both an FBS-containing and FBS-free culture system. In doing so, IGF-1 significantly enhanced PC3 and DU145 cell growth, compared to the untreated controls, as depicted in Figure 2. The effect was independent of whether the tumor cells were exposed to IGF-1 in cell culture medium containing 0, 2 or 10% FBS. Still, growth activity of DU145 and PC3 cells was lower when cultured in FBS free medium, compared to culturing the cells in medium enriched with 2 or 10% FBS. Therefore, subsequent experiments were carried out with tumor cells grown in 2% FBS. LNCaP and VCaP did not grow well in the presence of 0 and 2% FBS, and even in the presence of 10% FBS growth activity was only moderate, compared to DU145 and PC3 cells (Figure 2). Exposing LNCaP to IGF-1 did not result in a significant alteration of tumor growth, whereas a significant increase in VCaP cells was seen after 72 h IGF-1 incubation.

### 3.2. IGF-1 Alters Tumor Cell Adhesion and Elevates Chemotaxis

IGF-1 altered PC3 and DU145 tumor cell adhesion behavior. Adhesion to immobilized collagen significantly increased after 24 h IGF-1 stimulation. There was also a trend to increase attachment to fibronectin which, however, was not significant. Adhesion of DU145, but not PC3, was diminished significantly when the tumor cells were shortly pre-incubated with IGF-1 for 4 h (Figure 3A). Chemotaxis of both PC3 and DU145 was moderately diminished following a 4 h pre-stimulation with IGF-1, but considerably enhanced following a 24 h IGF pre-incubation (Figure 3B). Horizontal migration dynamics were additionally evaluated for DU145 and PC3 cells. Whether tumor cells were pre-incubated with IGF-1 for 4 (DU145 cells) or 24 h (DU145, PC3 cells), a distinct increase in motile activity was noted, compared to unstimulated controls (Figure 4). The FBS-content (2 versus 10%) did not influence the effect of IGF-1 on horizontal migration of DU145 or PC3 cells (Figure 4). LNCaP and VCaP cells did not show any chemotactic activity and were, therefore, not exposed to IGF-1.

### 3.3. Modification of Integrin Surface Expression

The basal integrin α and β surface expression levels on DU145 and PC3 cells show strong expression of the integrins α2, α3, α5, α6, αV, β1, and β4 on both cell lines, whereas α1 was only marginally detectable. Integrin member α4 was not expressed at all, β3 was slightly apparent on DU145 but not detectable on PC3 cells (Figure 5). The integrins α5 (moderately), αV, and β1 were detected on LNCaP. The subtypes α2, α3, α5, α6, αV, and β1 were detected on LNCaP cells (Figure 5).

IGF-1 induced significant alterations of the integrin α3, α5, αV, and β1 surface expression level on DU145 and PC3 cells, whereby the kind of alteration strongly depended on the IGF-1 incubation time. Concerning DU145 cells, integrin α3 was elevated after 4 h but diminished after 24 h under IGF-1. Distinct up-regulation of α5 became evident after 24 h IGF-1 incubation time. The integrin subtype αV considerably increased after 4 h and decreased thereafter (24 h), whereas β1 was up-regulated after 2 h but down-regulated after 4 and 24 h (Figure 6). Concerning PC3 cells, moderate differences were observed, compared to DU145 cells. Integrins α3 and α5 were diminished after 4 h. However, up-regulation of α5 became evident after 24 h. The integrin subtype αV considerably increased after both 4 and 24 h, whereas β1 was up-regulated after 2 h but down-regulated after 24 h (Figure 6).

Alterations of the integrins α5, αV, and β1 on LNCaP and VCAP cells induced by IGF-1 are shown in Figure 7. The αV subtype was reduced in both cell lines after 24 h. The integrins α5 and β1 were down-regulated on VCaP as well, but not on LNCaP cells.

Total integrin protein content was also analyzed in DU145 cells. Exposing DU145 cells to IGF-1 for 24 h significantly elevated integrin α3 and αV and reduced integrin α5, whereas β1 was not modified (Figure 8).

### 3.4. Cell Signaling Pathway

IGF-1 (24 h stimulus) acted on both cell growth and cell adhesion relevant signaling in DU145 cells. Akt (both total and phosphorylated) increased as well as phosphorylated Rictor (pRictor). In parallel, elevation of phosphorylated FAK (pFAK) was recorded. No response was seen with Raptor, ILK or total FAK (Figure 9, Appendix A). With respect to PC3 cells, IGF-1 (24 h stimulus) diminished integrin α3 and α5, and elevated αV. Integrin β1 was not altered (Figure 10, Appendix A). The signaling proteins pRictor, pRaptor, pAkt and pFAK were all up-regulated in PC3 cells following IGF-1 exposure (Figure 10, Appendix A).

### 3.5. Blocking Studies

Surface expression of integrin α3, α5, αV, or β1 was blocked by their respective function associated monoclonal antibodies and DU145, PC3, LNCaP, and VCaP growth was evaluated. Blocking integrin α3, αV, or β1 significantly suppressed tumor growth (DU145, PC3), whereas blocking α5 did not affect cell growth (shown for DU145), related to the controls (Figure 11). Blocking α5 moderately suppressed growth after 24 h in LNCaP cells. In contrast, VCaP cell growth was not affected by α5 blockage. αV blockade resulted in elevated growth activity in VCaP cells, while β1 blockade resulted in diminished growth activity in this cell line (Figure 11). Analysis of tumor chemotaxis (both DU145 and PC3) revealed significant diminishment in the presence of α3, αV, or β1 blocking antibodies, and significant elevation when integrin α5 was blocked (Figure 11).

Adhesion of DU145 and PC3 to collagen or fibronectin also correlated with the integrin surface level (Figure 12). Adhesion of DU145 to collagen was influenced by all blocking antibodies applied, with significant down-regulation by blocking α3, α5, or β1 and distinct elevation following αV blockade. Adhesion to fibronectin was diminished by anti-α5, -αV or -β1. The same effects were induced on PC3 adhesion to fibronectin. However, adhesion of PC3 to collagen was down-regulated by all anti-α3, -α5, or -αV antibodies (Figure 12).

Since protein analysis pointed to distinct alterations of integrins α3, α5, and αV and the integrin related signaling protein FAK in DU145 cells caused by IGF-1 (Figure 8 and Figure 9), alterations of DU145 cell growth or chemotactic behavior were also investigated using integrin α3, α5, αV and FAK knock-down, or integrin α3, α5 and αV knock-down, respectively. The protein profile under siRNA is shown in Figure 13A. Based on the MTT-assay, integrin α5 knock-down was associated with increased cell growth activity, whereas knocking down integrin α3, αV, or FAK resulted in a loss of tumor growth (Figure 13B). In contrast, chemotaxis was elevated when integrin α3 was knocked down, but diminished by α5 or αV specific siRNA (Figure 13C).

The effects of integrin blockade on tumor growth did not depend on IGF-1- or FBS-concentration. Figure 14A depicts DU145 cell growth data derived from cultures with 2 versus 10% FBS and with or without IGF-1-activation. The IGF-1-concentration did not influence integrin dependent alterations of DU145 chemotaxis, as shown in Figure 14B. Adhesion of DU145 cells to collagen and fibronectin in the presence of β1 function blocking antibodies was similarly blocked, whether tumor cells were grown in 2 or 10% FBS (Figure 14C).

In final experiments, cross-communication between Akt and integrin signaling was explored. Knocking down Akt was paralleled by increased expression of integrin α5 and loss of integrins α3 and αV (Figure 15 left, Appendix A). No effect was seen on β1 integrin and on FAK. We also evaluated whether FAK signaling influences mTOR signaling (Rictor, Raptor). Incubation of DU145 cells with the FAK-inhibitor defactinib distinctly diminished the expression of total and phosphorylated Rictor and Raptor after 72 h (Figure 15 right).

## 4. Discussion

Although the relevance of IGF-1 as a prognostic biomarker in prostate cancer patients has been demonstrated in several cohort studies, its mode of action remains unsettled. The present investigation employing prostate cancer cell lines in vitro point to a dual role of IGF-1. Aside from suppressing tumor growth, IGF-1 acted on cell adhesion and motility as well. Treating PC3 or DU145 cells for 24 h with IGF-1 resulted in a significant increase in adhesion to collagen and, to a minor extent, to fibronectin. This is important, since adhesion of tumor cells to collagen and fibronectin is necessary in forming a pre-metastatic niche, whereby elevated collagen production facilitates cancer cell colonization into the bone [13].

Elevated adhesion was accompanied by elevated migration activity, shown by both the Boyden chamber assay evaluating vertical migration towards a chemotactic stimulus and the wound healing assay focusing on horizontal cell movement. Our investigation provides evidence that IGF-1 acts on the expression level of integrin α and β subtypes, notably α3, α5, αV, and β1. Marelli et al. observed complete antagonization of IGF-1-induced migration of DU145 and PC3 cells following αVβ3 function blockade [14]. Since β3 integrin was not expressed on PC3 cells, as demonstrated by FACS analysis, we did not further deal with this subtype. In accordance with our experiments, we assume that αV rather than β3 contributed to the IGF-1-induced migration effects observed by Marelli et al. Still, the role of IGF-1 in regard to integrin expression is not straightforward. We have observed that the kind of integrin modulation strongly depends on the IGF-1 incubation time. DU145 surface expression of α3 and αV increased strongly after 4 h but was reduced after 24 h. The β1 subtype was elevated already after 2 h with subsequent down-regulation at 4 and 24 h, in contrast to α5, which was distinctly elevated after 24 h.

Slight differences in integrin modulation have also been noted between DU145 and PC3 cells. Particularly, αV was enhanced on PC3 cells after 4 or 24 h IGF-1 incubation. Hypothetically, this difference could explain why blocking αV surface expression correlated with an increased adhesion of DU145 cells but decreased PC3 adhesion to immobilized collagen.

The protein profile, investigated after 24 h IGF-1 incubation, indicates enhancement of α3 and αV and diminution of α5 in DU145 cells. This could point to IGF-1 causing a translocation of α3 and αV from the cell membrane into the cytoplasm and translocation of integrin α5 from inside the cell to the surface membrane. Integrin β1 was not distinctly modified (only a slight, insignificant reduction was apparent after 24 h IGF-1 treatment). Sayeed et al. have demonstrated that β1 integrin regulation by IGF-1R does not occur at the mRNA level [15]. This may explain why we did not observe intracellular β1 alterations at the 24 h time point. Based on their and our data, it may be assumed that β1 is shifted from the cancer cell surface only after 24 h. Protein expression at earlier time points has not been investigated. Thus, further modes of IGF-1 action should be considered, including β1-trafficking, as has been documented in an in vivo mouse xenograft model [16]. Since differences are seen in initial integrin expression levels, with β3 verified on DU145 but not on PC3 cells, integrin trafficking, if it does take place, must also differ. We have shown that integrin protein expression in response to IGF-1 does differ between DU145 and PC3 cells. Differences in the genetic pattern of different tumor types may possibly be involved in modulating particular integrin subtypes since DU145 harbors mutations in CDKN2A, RB1, and TP53, whereas PC3 harbors mutations in PTEN and TP53.

Time-dependent integrin alteration caused by IGF-1 has not been dealt with by others as yet. However, time-dependent expression kinetics of several integrin subtypes (α2, α5, αV, β1, β3) has been observed with the human umbilical vein cell line EA.hy926 and the hepatoblastoma cell line HepG2. The temporal sequence of integrin up- and down-regulation has been interpreted such that coordinated assembly and disassembly of these receptors might be necessary to allow coordinated cell migration [17,18]. Interaction between c-Met and β1-integrin receptors has recently been described [19]. This is notable, since stimulation of PC3 cells with IGF-1 induced a time-dependent phosphorylation of c-Met, reaching a maximum 18–24 h after IGF-1 addition [20].

Therefore, we assume that time-dependent alterations in the integrin subtype expression pattern, along with integrin translocation processes, are pivotal mechanisms accounting for how IGF-1 may force metastatic spread. Integrin alterations might further be coupled to cascaded downstream signaling, as observed in a HepG2 cell migration assay [21]. Our data point to an elevation of Akt/Rictor signaling under IGF. Since IGF-1 enhanced the levels of integrin α3 and αV while reducing α5, and Akt knock-down diminished α3 and αV but up-regulated integrin α5, cross-communication between these integrin subtypes and Akt seems likely. Interestingly, FAK also became activated by IGF-1, but was not altered following Akt knock-down, showing that Akt may not serve as an upstream modulator of FAK. This does not exclude the possibility that FAK may regulate IGF-1R activation and subsequently Akt, as the downstream effector. We did not investigate this issue, but FAK is known to interact with integrins and IGF-1R in osteogenic cells, which results in up-regulation of Akt phosphorylation [22].

The IGF-1-integrin axis is not only involved in tumor cell invasion but also in the control of tumor cell growth. Depending on the kind of integrin blockade and the type of integrin which was blocked, different modes of action were apparent. Surface blockade of integrin α5 did not cause any effects on cell growth but correlated with increased chemotaxis. In contrast, α5 knock-down was found to increase tumor growth but reduce chemotaxis. Blocking αV with its monoclonal antibody was associated with both diminished growth and migration, whereas siRNA knock-down enhanced tumor growth at the 48 h time point. Loss of the α3 subtype was associated with reduced growth, independent from the kind of integrin modulation. However, chemotaxis was diminished following α3 surface blockade, but elevated following α3 knock-down. We, therefore, conclude that IGF-1 alters integrin subtypes in a concerted and fine-tuned manner, which may finally accelerate both prostate cancer growth and invasion.

Since adhesion and chemotaxis were more strongly influenced by IGF-1 than was tumor cell growth, the action of IGF-1 on integrin subtype expression might be particularly important in forcing metastatic progression. Although this is speculative, we observed the most prominent alterations on the integrin αV subtype being excessively enhanced on the cell surface following 4 h IGF-1 incubation and considerably elevated in the cytoplasm after 24 h. Since αV expression strongly correlated with DU145 chemotaxis, as verified by both blocking and knocking down αV, this hypothesis seems plausible. In good accordance, αV has been found to be aberrantly expressed in bone-metastases of prostate adenocarcinomas and is suggested to play a candidate role in the bone dissemination of aggressive prostate cancer [8]. Recently, Ibrahim et al. reported that exposing breast cancer cells to IGF-1 may cause a shift from IGF-1 receptor substrate (IRS)-1 phosphorylation to IRS-2 phosphorylation, leading to activated cell migration [23]. In this context, expression of IRS-1 resulted in IGF-1-stimulated proliferation, but did not affect motility, whereas expression of IRS-2 enhanced IGF-1-stimulated motility but did not stimulate proliferation [24]. This is notable, since IRS-2 has already been linked to integrin αV as a cell migration regulator [25].

Our results also point to integrins as tumor growth modulators, being related to prior IGF-1 stimulation. The most prominent influence on tumor cell growth was exerted by the integrin subtype α3. Since α3 accumulates in the cell cytoplasm, at least after a 24 h IGF-1 stimulation, we postulate that this process might be relevant to the IGF-1 induced enhancement of the cell growth rate. Integrin α3 has been shown to activate colorectal cancer cell proliferation [26], and evaluation of papillary thyroid cancer tissues and cell lines indicate an influence of integrin α3 on cell cycling and autophagy [27].

The role of integrins in tumor growth has been demonstrated by others, whereby most studies concentrate on integrin β1. β1 activated IGF-1R signaling in DU145 and PC3 cells and promoted cell survival and proliferation [28,29]. The role of the β1 integrin as a trigger factor for IGF-1-mediated mitogenic and transforming activities has also been verified in androgen-sensitive LNCaP cells [30]. Signaling between the β1 integrin and IGF-1R is thought to enable Akt activation and further signaling pathways to start the mitotic cascade [31]. In our experimental model, the integrin β1 surface level was elevated at a very early time point and rapidly decreased under the influence of IGF-1 in both DU145 and PC3 cells. How far this regulatory mechanism may contribute to tumor cell mitosis is not clear. Nevertheless, all β1-related studies cited here have in common that integrin β1 served as an upstream mediator of IGF-1-triggered IGF-1R activation, which contrasts with our observations. We did not investigate whether β1 and further integrin family members, α3, α5 and αV, may contribute to IGF-1-related signaling, but rather investigated the inverse mechanism. Therefore, different integrin action modes should be considered, depending on their activation order.

IGF-1-stimulation did not alter LNCaP growth and only integrin αV was slightly reduced after 24 h incubation. We, therefore, assume that the IGF-1-integrin-interaction seen in the androgen-resistant DU145 and PC3 cells is not highly relevant in the androgen-sensitive LNCaP cell line. Integrin blockage resulted in no significant difference in cell growth after 72 h. However, in contrast to LNCaP, IGF-1 did activate VCaP tumor cell growth and caused diminishment of the integrins α5, αV, and β1, pointing to differences between the LNCaP and VCaP cell lines. The IGF ligand-neutralizing antibody xentuzumab has been shown to alter the proliferative activity of VCaP but not of PTEN-null LNCaP cells. The authors assumed that PTEN may be involved in IGF-1 triggered cell proliferation [32]. Considering that PTEN-null PC3 cells responded well to IGF-1 treatment in terms of cell growth and integrin regulation, this postulate may not hold true for our model. However, IGF-1R was strongly expressed on VCaP but not on LNCaP cells, which may at least partially account for the differences observed. Since the response of the androgen-sensitive LNCaP and VCaP cell lines to IGF-1-stimulation differs, androgen-sensitivity per se does not appear to be a characteristic that uniformly influences the response to IGF-1-stimulation.

Overall, an inconsistent picture is presented here. IGF-1 driven cross-communication between Akt/mTOR and FAK-integrin signaling has been demonstrated in the androgen-resistant prostate cancer cell lines DU145 and PC3. Since integrin β1 was altered similarly by IGF-1 and shown to down-regulate growth and invasion of both cell lines, functional blockade of β1 might be an option to treat prostate cancer once castration-resistance has been established. Differences between the androgen-resistant and the androgen-sensitive cells became obvious. LNCaP cells did not respond to IGF-1 in terms of growth activation and integrin expression (excepting a slight loss of αV). VCaP differed in a way that blockade of integrin αV did not diminish cell growth as seen with DU145 and PC3 but was rather associated with an accelerated cell growth. Presumably, IGF-1 driven Akt-integrin cross-talk might (at least partially) depend on androgen related signaling. The initial integrin equipment which varied considerably between DU145/PC3, LNCaP, and VCaP, may also be a crucial factor determining the kind of Akt-integrin communication.

## 5. Conclusions

Evidence is provided that IGF-1 modulates the integrin α3, α5, αV and β1 expression pattern and distribution, which is associated with growth, adhesion and migration of DU145 and PC3 prostate cancer cells in vitro. Although all of these integrin subtypes were involved in both tumor growth and chemotaxis, we speculate that the αV subtype may particularly contribute to the elevation of tumor cell migration, whereas integrin α3 may predominantly regulate tumor cell growth. No clearcut differences in the response to IGF-1 stimulation were apparent in regard to androgen-sensitive cells compared to androgen-insensitive cells. Certainly, further investigation is required before the role of integrin targeting in prostate cancer management can be more clearly delineated.

## Figures and Tables

**Figure 1 cancers-14-00363-f001:**
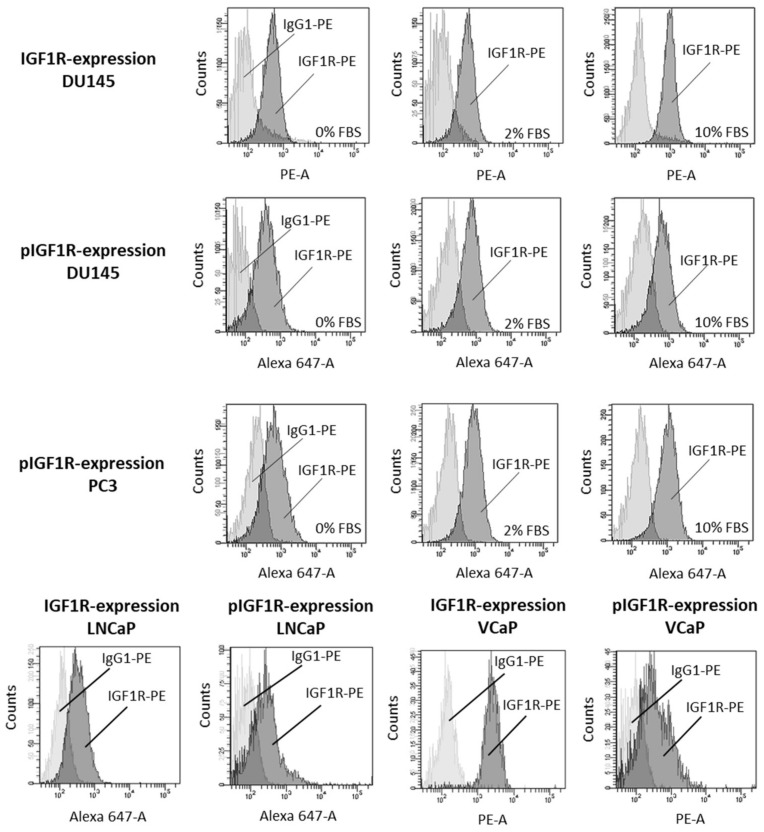
IGF-1R (IGF1R) and pIGF-1R (pIGF1R) expression levels in DU145, PC3, LNCaP and VCaP cells. Light gray: isotype control; dark gray: specific fluorescence. Flow cytometry curves are representative for one of three experiments.

**Figure 2 cancers-14-00363-f002:**
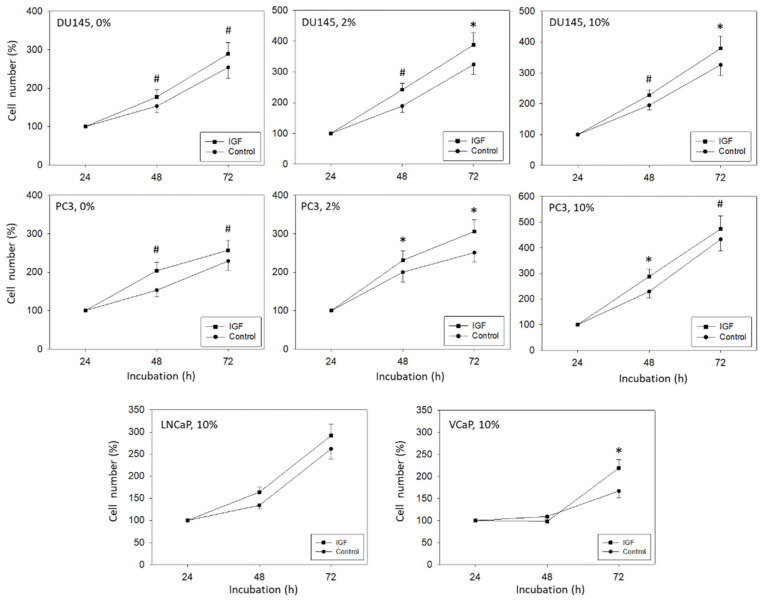
DU145, PC3, LNCaP, and VCaP cell growth in response to IGF-1. DU145 and PC3 cells were grown in FBS free medium (0%), in medium containing 2% FBS (2%) or 10% FBS (10%). LNCaP and VCaP were cultivated in the presence of 10% FBS. Untreated cells served as controls. Cell number was evaluated after 24, 48, and 72 h by the MTT assay. Error bars indicate standard deviation. Experiments were repeated five times. One representative experiment is shown. * indicates *p* < 0.05, # indicates *p* < 0.01.

**Figure 3 cancers-14-00363-f003:**
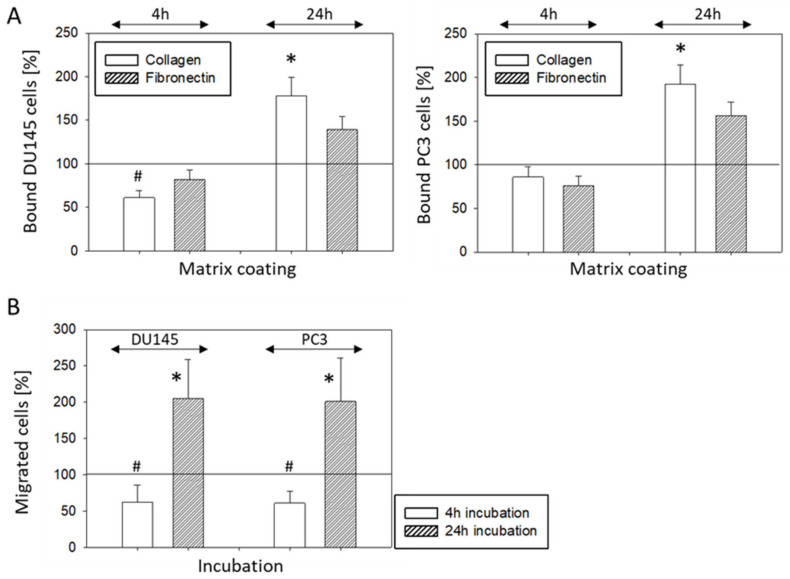
(**A**) Adhesion of DU145 and PC3 cells to immobilized collagen or fibronectin after stimulation with IGF-1 for 4 or 24 h. Mean number of adherent tumor cells from five fields. * indicates significant up-regulation to untreated control, # indicates significant down-regulation to untreated control (*n* = 4). (**B**) Chemotactic movement of DU145 and PC3 cells after stimulation with IGF-1 for 4 or 24 h. Controls remained untreated. Mean number of tumor cells crawling beneath the filter membrane. * indicates significant up-regulation to untreated control, # indicates significant down-regulation to untreated control (*n* = 3). All values are related to untreated controls set to 100%.

**Figure 4 cancers-14-00363-f004:**
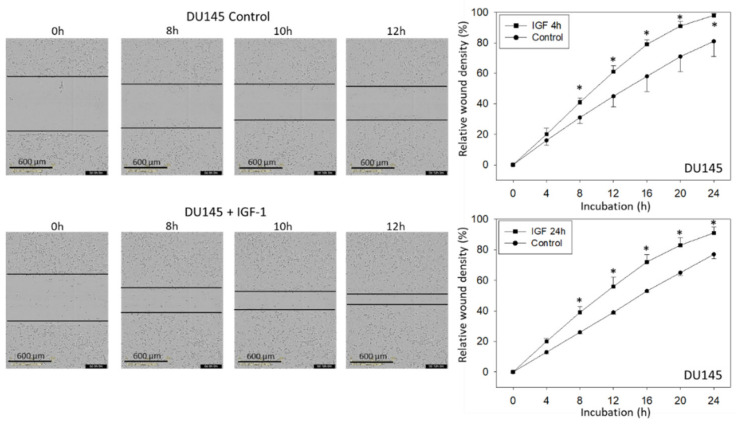
Motile crawling of DU145 and PC3 cells treated with IGF-1 (versus controls). The upper left side shows photomicrographs of the DU145 scratch assay taken after 0 (start), 8, 10, and 12 h (related to 4 h IGF-1 pre-stimulation). The upper right side shows the results from the quantitative calculation expressed as relative wound density (4 and 24 h pre-stimulation). DU145 were grown in 2% FBS. Lower panels: Relative wound density of DU145 cells following 4 or 24 h IGF-1 pre-stimulation and cultivation in 10% FBS, and relative wound density of PC3 cells stimulated with IGF-1 for 24 h and cultured in 2 or 10% FBS. Mean values of three experiments are shown. * indicates significant up-regulation compared to untreated controls (*n* = 3).

**Figure 5 cancers-14-00363-f005:**
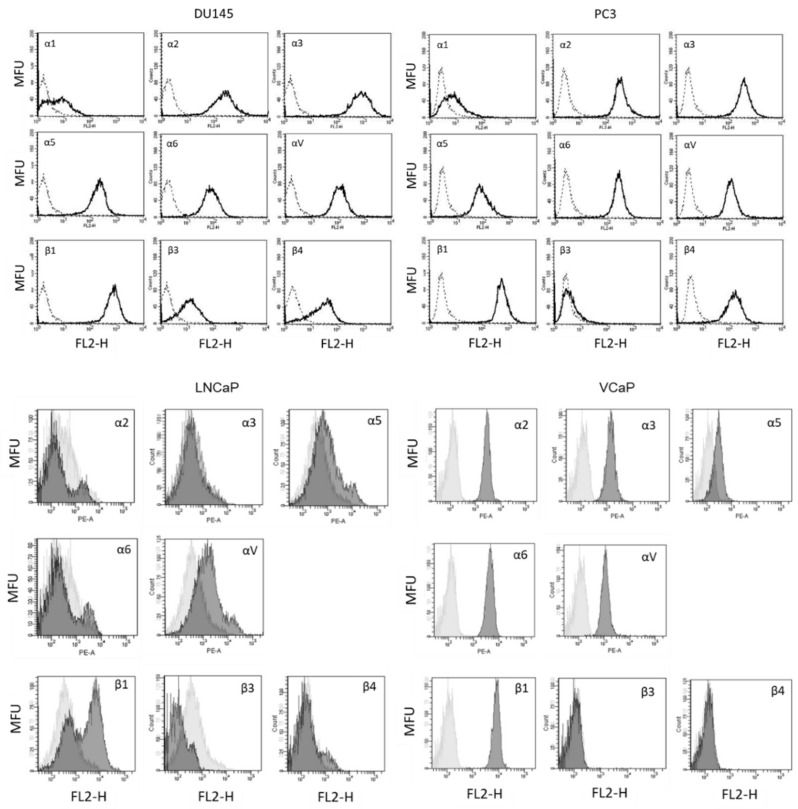
Surface expression of integrin α and β subtypes on DU145, PC3, LNCaP, and VCaP cells. Counts indicate cell number; fluorescence is expressed by mean fluorescence units (MFU). One representative of three separate experiments is shown. Solid line = specific fluorescence, dotted line = isotype control.

**Figure 6 cancers-14-00363-f006:**
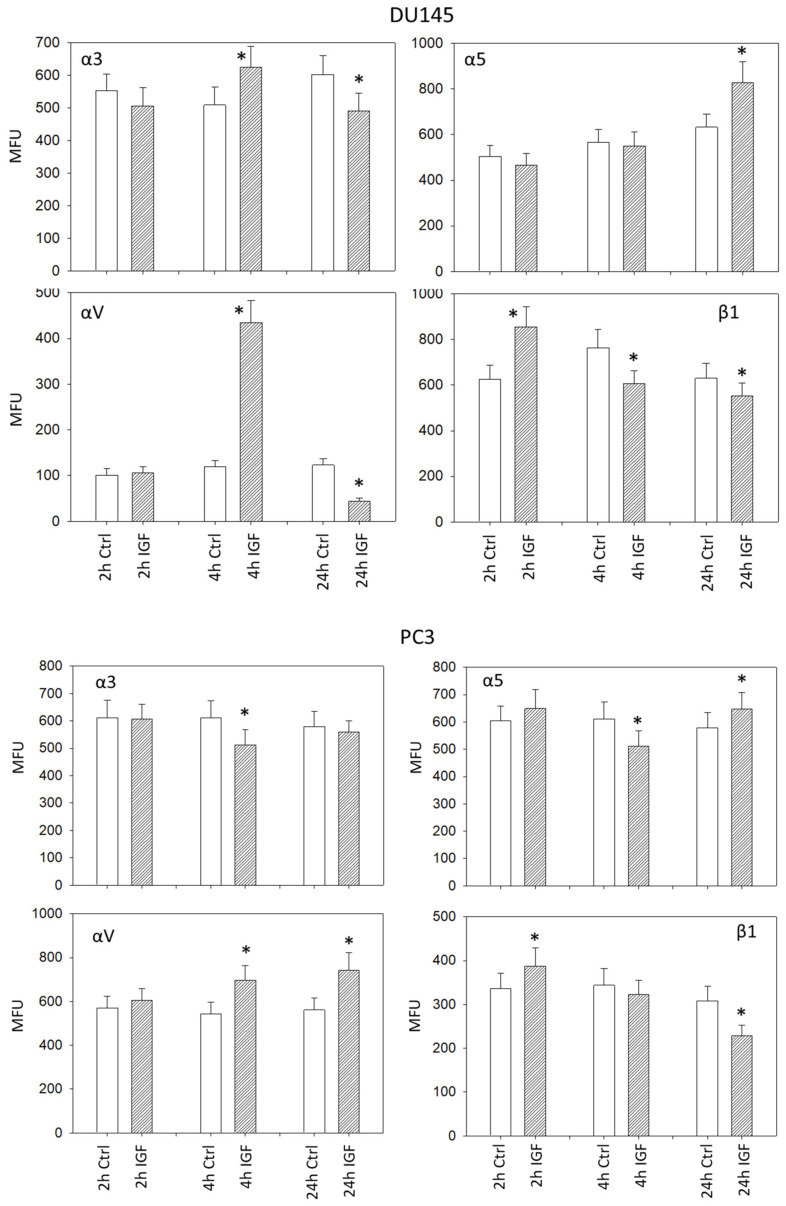
IGF-1 stimulated surface expression of the integrins α3, α5, αV, and β1 on DU145 and PC3 cells, evaluated after 2, 4 and 24 h. All values are related to untreated controls. MFU: Mean fluorescence units. * indicates significant difference to controls (*n* = 3).

**Figure 7 cancers-14-00363-f007:**
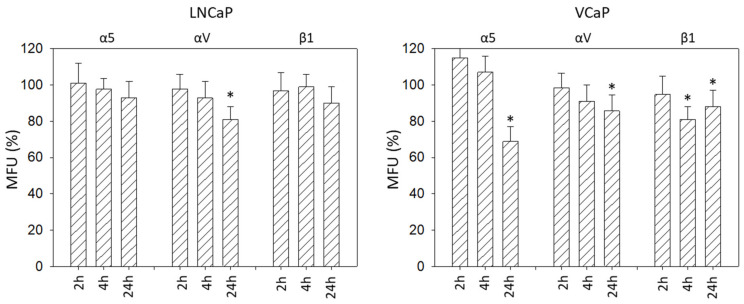
IGF-1 stimulated surface expression of the integrins α5, αV, and β1 on LNCaP and VCaP cells, evaluated after 2, 4, and 24 h. All values are related to untreated controls set to 100%. MFU: Mean fluorescence units. * indicates significant difference to controls (*n* = 3).

**Figure 8 cancers-14-00363-f008:**
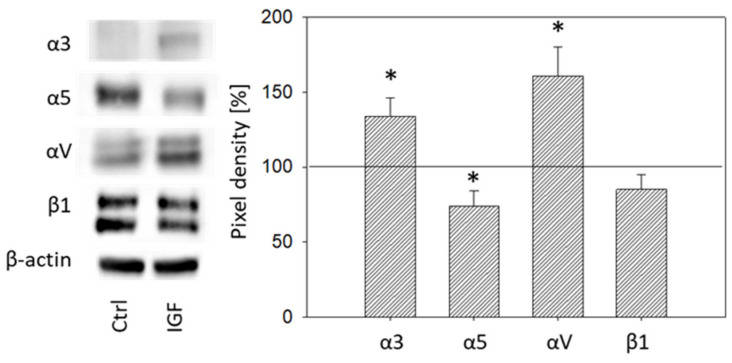
Left side: Protein profile of integrins α3, α5, αV, and β1. DU145 cells were either stimulated with IGF-1 for 24 h or exposed to culture medium without IGF-1 (Ctrl). One representative of three separate experiments is shown. Each protein analysis was accompanied by a β-actin loading control. One representative internal control is shown. Right side: Pixel density analysis of the protein expression level of DU145 cells stimulated with IGF. The ratio of protein intensity/β-actin intensity is expressed as percentage of controls, indicated by line at 100%. * indicates significant difference to controls.

**Figure 9 cancers-14-00363-f009:**
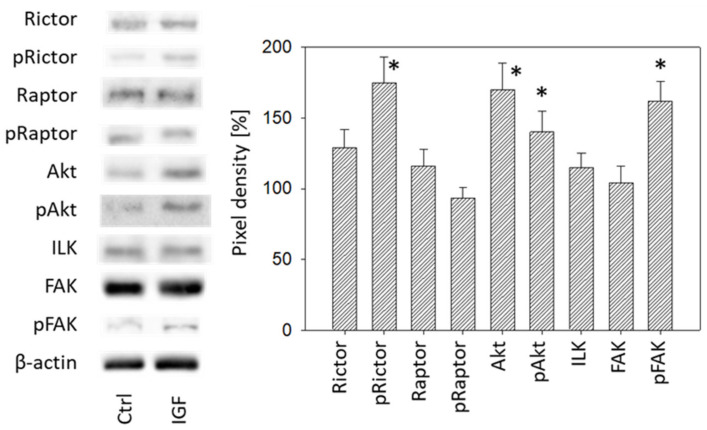
Left: Protein profile of cell signaling proteins. DU145 cells were either stimulated with IGF for 24 h or received culture medium without IGF (Ctrl). One representative of three separate experiments is shown. Each protein analysis was accompanied by a β-actin loading control. One representative internal control is shown. Right: Pixel density analysis of the protein expression level of DU145 cells stimulated with IGF. The ratio of protein intensity/β-actin intensity is expressed as percentage of controls set to 100%. * indicates significant difference to controls.

**Figure 10 cancers-14-00363-f010:**
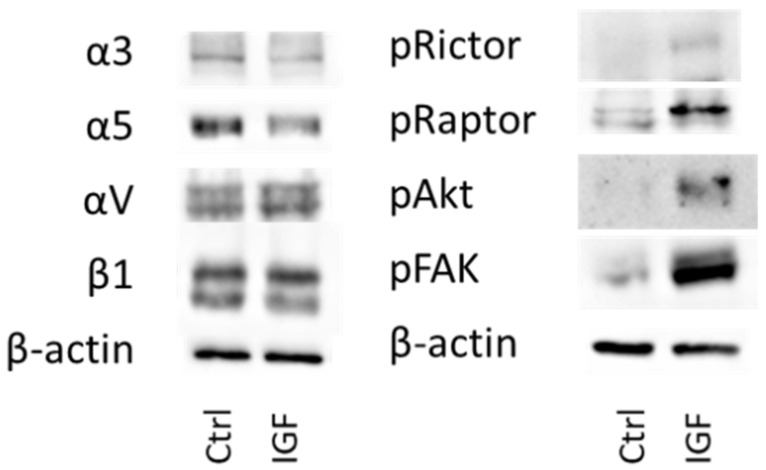
Left: protein profile of integrins α3, α5, αV, and β1 in PC3 cells. Right: protein profile of cell signaling proteins in PC3 cells. DU145 or PC3 cells were either stimulated with IGF for 24 h or received culture medium without IGF (Ctrl). One representative of three separate experiments is shown. Each protein analysis was accompanied by a β-actin loading control. One representative internal control is shown.

**Figure 11 cancers-14-00363-f011:**
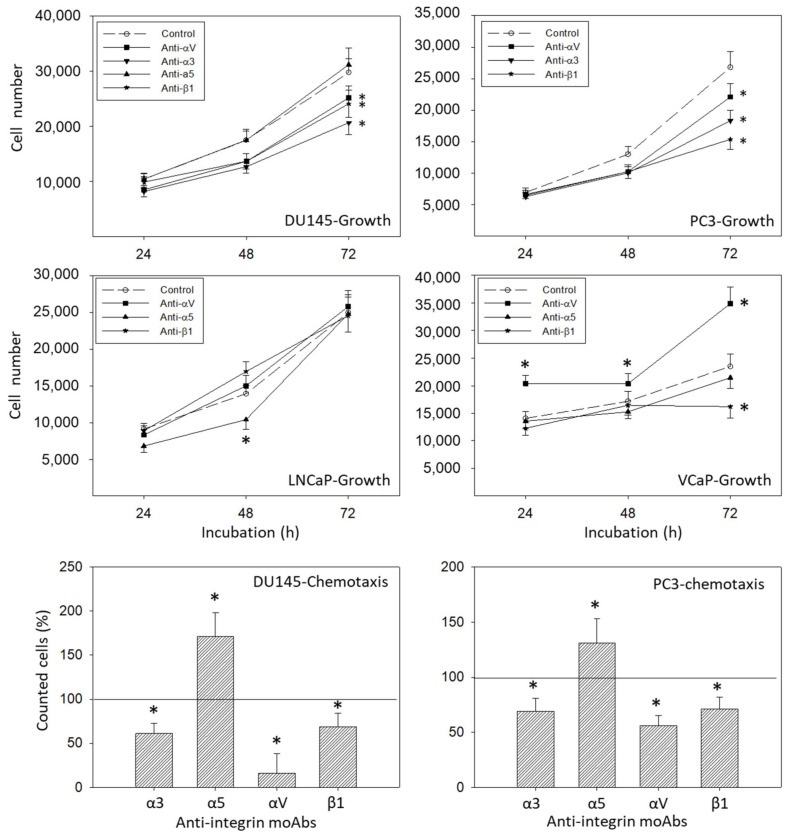
Upper panels: cell growth dynamics of DU145, PC3, LNCaP, and VCaP cells after blockade through monoclonal antibodies against α3, α5, αV, or β1. Controls remained untreated. * indicates significant difference to untreated controls (*n* = 4). Lower panels: chemotactic movement of DU145 and PC3 cells after blockade through monoclonal antibodies (specific blockade against α3, α5, αV, or β1). Controls remained untreated. All values are related to the untreated controls set to 100%. * indicates significant difference (*n* = 3).

**Figure 12 cancers-14-00363-f012:**
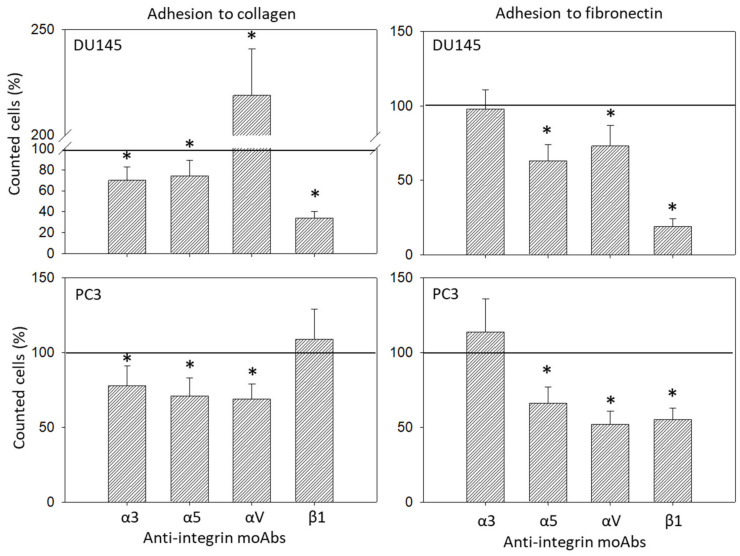
Adhesion modulation of DU145 and PC3 through monoclonal antibodies against α3, α5, αV, or β1 (left: adhesion to collagen; right: adhesion to fibronectin). Controls remained untreated. * indicates significant difference to untreated controls (*n* = 3).

**Figure 13 cancers-14-00363-f013:**
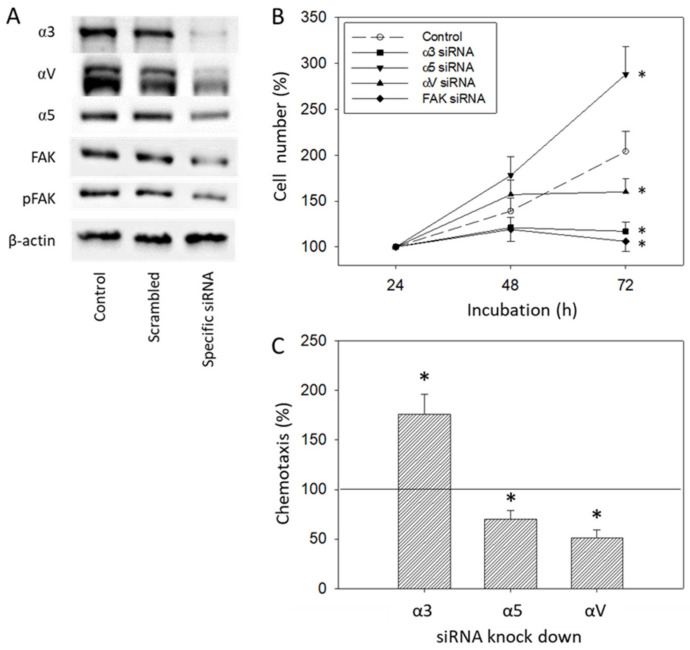
(**A**) Protein expression level following siRNA transfection (untreated control versus scrambled siRNA versus specific siRNA). Each protein analysis was accompanied by a β-actin loading control. One representative internal control is shown. (**B**) Cell growth of DU145 treated with an integrin α3, α5, αV, or FAK specific siRNA, evaluated by the MTT-assay. One representative of three separate experiments is shown. (**C**) Chemotactic movement of DU145 cells after knocking down integrin α3, α5, or αV. All values are related to the scrambled controls set to 100%. * indicates significant difference to controls (*n* = 3).

**Figure 14 cancers-14-00363-f014:**
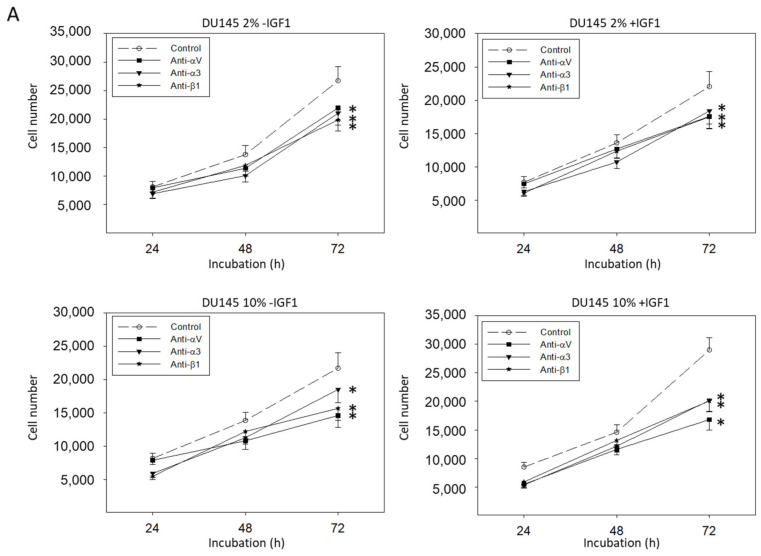
(**A**) DU145 cell number in response to integrin blockade. Tumor cells were grown in 2 or 10% FBS and in the presence or without IGF-1. Untreated cells served as controls. Cell number was evaluated after 24, 48, and 72 h by the MTT assay. (**B**) Chemotactic movement of DU145 cells after blockade through monoclonal antibodies (specific blockade against α3, αV, or β1) with or without IGF-1 activation. (**C**) Adhesion modulation of DU145 through integrin β1 blockade in the presence of 2 versus 10% FBS. (**B**,**C**) are related to the non-blocked controls set to 100%. Error bars indicate standard deviation. Experiments were repeated three times. One representative experiment is shown. * indicates *p* < 0.05.

**Figure 15 cancers-14-00363-f015:**
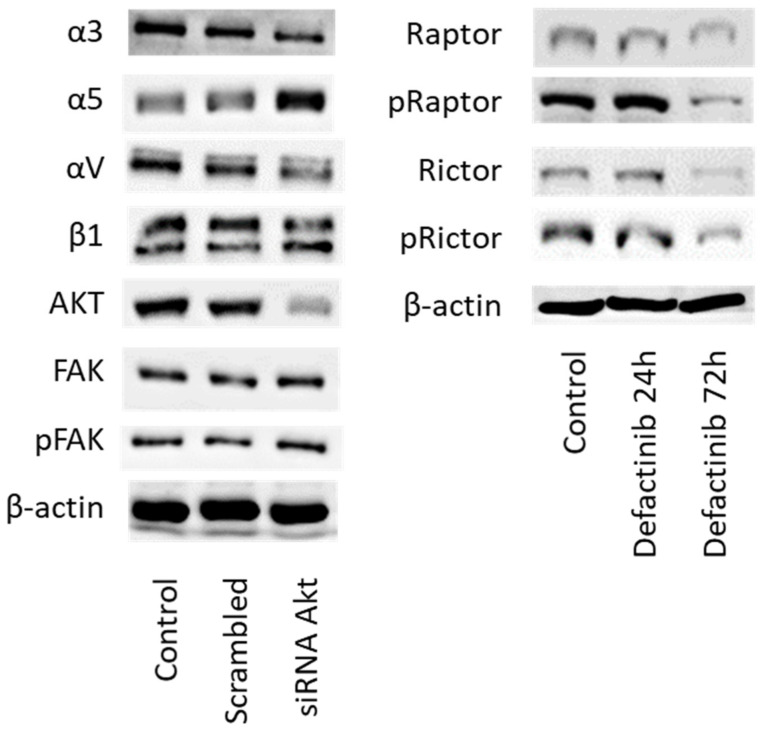
(**Left**) Integrin subtype and FAK expression level in DU145 cells following treatment with Akt specific siRNA (versus untreated controls or scrambled siRNA). (**Right**) mTOR signaling in DU145 cells following defactinib treatment for 24 and 72 h. Controls remained untreated. One representative of three separate experiments is shown. Each protein analysis was accompanied by a β-actin loading control. One representative internal control is shown.

## Data Availability

Data sharing not applicable.

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
