# Peer review of "Insulin-like Growth Factor-1 Influences Prostate Cancer Cell Growth and Invasion through an Integrin α3, α5, αV, and β1 Dependent Mechanism"

_cancers, 2022, doi:10.3390/cancers14020363_

Round 1

Reviewer 1 Report

The authors have performed a number of additional experiments that have been suggested by the reviewers, including experiments with androgen-sensitive cell lines, so that the revised version of the manuscript can be - to my mind - published in its present form.

Author Response

We are thankful for the response of the referee. 

Reviewer 2 Report

The manuscript now more nicely describes, and even highlights, the significant differences in responses to growth factors like IGF1, and in articular integrin signaling, between the most commonly used "classic" prostate cancer cell lines.. Essentially, it points to the fact that these cell lines differ significantly, and may or may not represent certain PrCa subtypes (or progression stages) in the clinics. The addition of androgen-dependent (or t least responsive) cell lines to a slightly larger picture was, therefore, necessary and certainly beneficial - even if this may not have changed which cell lines respond most strongly to IGF1; and the corresponding response of the integrins. 

Although there cannot be put some concrete criticism what exactly is missing - one still has to wonder, however: what precisely is the relevance of the data, collected across 4 cell lines, for prostate cancer progression, therapy, maybe diagnostics - and where exactly is the novelty? These are mainly descriptive data, across very different cell lines, that are difficult to connect. I think the authors should highlight - in their discussion section - what they consider the most relevant findings, and the most important novelty? Otherwise, this article almost leaves no traces in the reader's mind ... what exactly have we learned about PrCa? What do the 4 cell lines tell us?  

Author Response

We agree that our data are not homogenous and the role of integrins may be more complex than we initially thought. We have now included a final para to summarize the most important findings and hope that the referee might be satisfied by the new version of the manuscript. Discussion, last para, reads:   

"Overall, an inconsistent picture is presented here. IGF-1 driven cross-communication between Akt/mTOR and FAK-integrin signaling has been demonstrated in the androgen-resistant prostate cancer cell lines DU145 and PC3. Since integrin β1 was altered similarly by IGF-1 and shown to down-regulate growth and invasion of both cell lines, functional blockade of β1 might be an option to treat prostate cancer once castration-resistance has been established. Differences between the androgen-resistant and the androgen-sensitive cells became obvious. LNCaP cells did not respond to IGF-1 in terms of growth activation and integrin expression (excepting a slight loss of αV). VCaP differed in a way that blockade of integrin αV did not diminish cell growth as seen with DU145 and PC3 but was rather associated with an accelerated cell growth. Presumably, IGF-1 driven Akt-integrin cross-talk might (at least partially) depend on androgen related signaling. The initial integrin equipment which varied considerably between DU145/PC3, LNCaP, and VCaP, may also be a crucial factor determining the kind of Akt-integrin communication".

This manuscript is a resubmission of an earlier submission. The following is a list of the peer review reports and author responses from that submission.

Round 1

Reviewer 1 Report

the main problem with this manuscript may relate to the unclear scientific significance of the experiments, which are largely performed based on a single PrCa cell line - DU145. DU145 and PC3 are both considered "outlier" cell lines, due to the fact that they have acquired a complete functional loss of androgen-receptor expression, which is rare in patients that undergo castration-resistant progression. It is therefore also unclear if the mechanism described, which are largely related to the downstream effects of IGF1 stimulation, on expression and functions of integrins and possibly the FAK signaling pathway. It remains unclear in this manuscript to which degree the finding presented are actually related to clinical issues, leading to the generation of castration-resistant cancers, and resistance to anti-androgen therapies. Simultaneously, there have been studies related to IGF signaling in prostate cancers for at least the last 20 years (over 110 papers), some of these showing that there is a correlation with expression and activity of integrins and IGF (PMID: 11313980, PMID: 16061650, PMID: 16465378, PMID: 20127733, etc...). It is true that some of these focus in ITGB1, but other integrins are covered as well. Thus, there is a bit of a lack of scientific novelty, especially in relation to potential clinical consequences or mechanisms of action, downstream of IGF and upstream of the integrins. Plus, although this is stated in the title, there is not much focus on the FAK signaling pathway, which would be a very good candidate to be included in this study. For example, there are many FAK inhibitors out there, available from the usual providers of small-molecule inhibitors for research purposes (Y-11, Defactinib, or PF-431396). 

Furthermore, there is a lack of correlation of the findings presented here with additional data, out there in diverse databases - for example, the authors could easily explore the relevance of altered mRNA or protein expression in clinical prostate cancer samples, such as by mining resources like cBioportal, or protein atlas; and many others. This would allow the discussion to focus more on the clinical and functional relevance of the findings described here, and how they may relate to cancer progression - especially towards metastasis and castration-resistant prostate cancer. 

The study would benefit a lot from addition of at least 1 or 2 androgen-receptor-positive cell lines, such as VCAP or LNCAP. Although these cell lines are not invasive, and some studies like invasion assays are not suitable with them - they will most certainly respond to IGF, and they express integrins that may be affected in a totally different fashion compared to the rather artificial (or poorly representative) cell lines PC3 and DU145. This would definitely increase the scientific merit of the manuscript, its soundness, and interest to the readers. It would probably also give the discussion a more interesting twist and almost certainly affect the main message generated. 

Manuscripts that are largely (like here) or even entirely based on a single cell line, and particularly if this is a cell line that may not be the physiologically most representative model system for the respective disease, are always to be considered flawed. I believe this is to some degree the case here. It would most definitely benefit the interpretation of results if additional cell lines would be added. These can still represent castration-resistant PrCa, but different mechanisms to interfere with androgen signaling and dependency in PrCa will almost certainly have differential effects on potential "rescue" pathways like insulin signaling. There is a large body of literature related to the potential connections of AR signaling with oncogenic pathways such as PI3K, AKT, mTOR; and IGF signaling. Vice versa, there is also a (smaller, but still significant) body of literature available that addresses the potential connection between these pathways, integrin activities, and progression, invasion, or metastasis. Thus, adding more cell lines into the "big picture" would clearly benefit the discussion and the scientific impact of the manuscript. 

Furthermore, there would be more solid and physiologically more relevant data, if various cell culture modalities would be used: this could include 3D cultures, or at least culture of cells embedded in ECM preparations such as Matrigel, collagen type I, and others which mimic the extracellular matrix. Furthermore, and especially for investigating FAK signaling and integrin functions, the co-culture with cancer-associated fibroblasts would be extremely beneficial. This can very significantly affect the functionality of FAK signaling and integrin ligand/receptor interactions, but none of it is covered here even remotely. 

Author Response

Comment 1: There is not much focus on the FAK signaling pathway, which would be a very good candidate to be included in this study. For example, there are many FAK inhibitors out there, available from the usual providers of small-molecule inhibitors for research purposes (Y-11, Defactinib, or PF-431396).

Our answer: To put more focus on the FAK signaling pathway we did carry out additional studies on DU145 cells treated with defactinib. Materials and Methods, section “Blocking studies and siRNA knock down” now reads (line 197): “In addition, DU145 cells were treated with 10 µM of the FAK-inhibitor defactinib (Biozol, Eching, Germany) for 24 and 72 h. The mTOR related signaling proteins, Rictor and Raptor, both total and phosphorylated, were then evaluated with Western blotting. ”. Data are shown in figure 15. Results, section 3.5 Blocking studies, now reads (line 431): “We also evaluated whether FAK signaling influences mTOR signaling (Rictor, Raptor). Incubation of DU145 cells with the FAK-inhibitor defactinib distinctly diminished the expression of total and phosphorylated Rictor and Raptor after 72 h (figure 15 right)”.

Comment 2: The authors could easily explore the relevance of altered mRNA or protein expression in clinical prostate cancer samples, such as by mining resources like cBioportal, or protein atlas; and many others.

Our answer: We checked the cBioportal and TCGA databases. From the 489 clinical prostate cancer samples, co-mutations of the IGF1 and integrin α5 genes were found, but only in 1% of patients. This small number does not allow calculation of the probability of overall or progression free survival. We, therefore, regret that we cannot say anything about the relevance of altered mRNA or protein expression in clinical prostate cancer samples.

Comment 3: The study would benefit a lot from addition of at least 1 or 2 androgen-receptor-positive cell lines, such as VCAP or LNCAP. This would definitely increase the scientific merit of the manuscript, its soundness, and interest to the readers. Adding more cell lines into the "big picture" would clearly benefit the discussion and the scientific impact of the manuscript.

Our answer: We have now additionally dealt with LNCaP and VCaP cell lines. Methods have been changed accordingly. New data sets are presented including IGF1R expression in LNCaP/VCaP (figure 1), LNCaP/VCaP cell growth in the presence of IGF1 (figure 2), integrin expression on LNCaP/VCaP (figure 5), IGF1 stimulated integrin expression on LNCaP/VCaP (figure 7) and LNCaP/VCaP cell growth blockade by integrins (figure 11).

The Introduction now reads (line 69): “Using human androgen-independent prostate cancer cell lines (DU-145, PC3) as well as androgen-sensitive LNCaP and VCaP cells, the present study was designed to evaluate how IGF1 is involved ….”.

Methods, Cell lines, now reads (line 76): “The human prostate tumor cell lines DU145, PC3, and LNCaP were obtained from DSMZ (Braunschweig, Germany). VCaP cell lines were obtained from the Department of Urology and Pediatric Urology, Saarland University, Homburg/Saar, Germany. DU145, PC3, and LNCaP tumour cell lines were grown in …. (line 83) VCaP were grown in DMEM medium, supplemented with FBS, 1% penicillin/streptomycin, 2% GlutaMAX and 1% sodium pyruvate (all: Gibco/Invitrogen)”.

Results, 3.1 IGF1 activates tumor cell growth, now reads (line 224): “IGF1R expression (total and phosphorylated) was verified in the androgen-sensitive cell lines LNCaP and VCaP as well (figure 1) ….. (line 234) LNCaP and VCaP did not grow well in the presence of 0% and 2% FBS, and even in the presence of 10% FBS growth activity was only moderate, compared to DU145 and PC3 cells (figure 2). Exposing LNCaP to IGF1 did not result in a significant alteration of tumor growth, whereas a significant increase in VCaP cells was seen after 72 h IGF1 incubation.

Results, 3.2 IGF1 alters tumor cell adhesion and elevates chemotaxis, now reads (line 266): “LNCaP and VCaP did not show any chemotactic activity and were, therefore, not exposed to IGF1.

Results, 3.3 Modification of integrin surface expression, now reads (line 300): “The integrins α5 (moderately), αV, and β1 were detected on LNCaP. The subtypes α2, α3, α5, α6, αV, and β1 were detected on LNCaP cells (figure 5). …. (line 331) Alterations of the integrins α5, αV, and β1 on LNCaP and VCAP cells induced by IGF1 are shown in figure 7. The αV subtype was reduced in both cell lines after 24 h. The integrins α5 and β1 were down-regulated on VCaP as well, but not on LNCaP cells.”.

Results, 3.5 Blocking studies, now reads: “Surface expression of integrin α3, α5, αV, or β1 was blocked by their respective function associated monoclonal antibodies (line 382) and DU145, PC3, LNCaP, and VCaP growth was evaluated. Blocking integrin α3, αV, or β1 significantly suppressed tumor growth (DU145, PC3), whereas blocking α5 did not affect cell growth (shown for DU145), related to the controls (figure 11). (line 385) Blocking α5 moderately suppressed growth after 24 h in LNCaP cells. In contrast, VCaP cell growth was not affected by α5 blockage. αV blockade resulted in elevated growth activity in VCaP cells, while β1 blockade resulted in diminished growth activity in this cell line (figure 11)”.

LNCaP and VCaP data are now discussed (line 588). “IGF1-stimulation did not alter LNCaP growth and only integrin αV was slightly reduced after 24 h incubation. We, therefore, assume that the IGF1-integrin-interaction seen in the androgen-resistant DU145 and PC3 cells is not highly relevant in the androgen-sensitive LNCaP cell line. Integrin blockage resulted in no significant difference in cell growth after 72 h. However, in contrast to LNCaP, IGF1 did activate VCaP tumor cell growth and caused diminishment of the integrins α5, αV, and β1, pointing to differences between the LNCaP and VCaP cell lines. The IGF ligand-neutralizing antibody xentuzumab has been shown to alter the proliferative activity of VCaP but not of PTEN-null LNCaP cells. The authors assumed that PTEN may be involved in IGF1 triggered cell proliferation [32]. Considering that PTEN-null PC3 cells responded well to IGF1 treatment in terms of cell growth and integrin regulation, this postulate may not hold true for our model. However, IGF1R was strongly expressed on VCaP but not on LNCaP cells, which may at least partially account for the differences observed. Since the response of the androgen-sensitive LNCaP and VCaP cell lines to IGF1-stimulation differs, androgen-sensitivity per se does not appear to be a characteristic that uniformly influences the response to IGF1-stimulation.

Comment 4: There would be more solid and physiologically more relevant data, if various cell culture modalities would be used: this could include 3D cultures, or at least culture of cells embedded in ECM preparations such as Matrigel, collagen type I, and others which mimic the extracellular matrix. Furthermore, and especially for investigating FAK signaling and integrin functions, the co-culture with cancer-associated fibroblasts would be extremely beneficial.

Our answer: We agree that more physiologically accordant cell culture models are important to obtain relevant data. The present study was designed as an overview and precursor to work involving such models. To increase the scope of the overview (see comments 1 and 3), we added additional tumor sublines including another androgen-resistant cell line (PC3) and two androgen-sensitive cell lines (LNCaP, VCaP). We have also gone into more detail in regard to  the influence of FAK-blockade on cell growth and chemotaxis as well as on IGFR1, Rictor, Raptor and Akt expression. We also evaluated the influence of IGF1 under integrin blockade (cell growth and chemotaxis). Finally, studies were carried out to compare tumor cell behavior in the presence of 10% versus 2% IGF (please see our answer to comment 2 of referee 3). Work directly involving 3D models and/or co-culture experiments would have considerably overloaded the manuscript and cannot be performed in a manageable time frame. In addition, 3D models and co-culture experiments involve tumor-stroma interaction, which portends a great increase in complexity. 3D models and co-culture experiments are indeed a next step and based on the present study we intend to venture into this area of research.  

Reviewer 2 Report

I have no comments.

Author Response

Comment 1: I have no comments.

Our answer: We are very thankful for this positive response. 

Reviewer 3 Report

In their work the authors investigated the influence of IGF1 in prostate cancer cell proliferation and migration and the role of various integrin cell adhesion molecules. Though the study is interesting per se, it was somehow unclear why the authors used two AR negative prostate cancer cell lines in their study, PC-3 and DU-145, which are not representative for the majority of prostate cancers, being AR positive and AR sensitive. Thus, confirming the most important findings in an AR positive cell line is strongly recommended in terms of finding new drug targets. In addition, since the authors concluded from their findings that there might be a cross-communication between integrin subtypes and AKT, it was not clear why the authors restricted their experiments on integrin expression to the PTEN positive DU-145 cell line. It is known that loss of PTEN as it occurs in PC-3 cells and subsequent activation of AKT frequently occurs in prostate cancer. Hence it would have been interesting to confirm the findings in PTEN negative PC-3 cells, too.

Besides, there were some minor issues which are listed below:

  • Cell growth assays were performed in medium with 0%, 2% and 10 % FBS. The authors speculated that FBS influences or masks the effect of IGF1, which is most probably the case. Have the authors measured how much IGF1 is in the FBS?
  • It would have been interesting to see the effect of neutralizing antibodies to integrins on cell growth in 0%, 2% and 10% FBS.
  • With regard to the experiments with neutralizing antibodies it was not clear if they were done in the absence or presence of IGF-1. To my mind, the neutralizing antibodies should have been used also in the adhesion experiments to give an information on the role of the different integrins in adhesion to collagen and fibronectin in prostate cancer cells.
  • To my mind, Figure 4 could be omitted since the scratch wound assay does not really give any additional valuable information and – besides – has only been done in DU-145 cells.
  • Flow cytometry and Western blotting revealed different results with regard to the expression of integrin subtypes. Which conditions were used for cell lysis in Western blot experiments? Do the lysates really only contain intracellular integrins or rather a mixture of intra- and extracellular integrins? Could the discrepant results be a consequence of using different antibody clones for Western blotting and flow cytometry?
  • 9 and Fig. 10: As far as I understood these experiments were conducted in the absence of IGF1. To my mind, however, it would be rather interesting to see the effects in the absence and presence of IGF1.
  • In the discussion, the authors claimed that IGF1 might cause a translocation of a3 and aV from the cell membrane into the cytoplasm and a translocation of a5 from inside the cell to the surface membrane. Would it be possible to show this translocation by immunofluorescence?

Author Response

Comment 1: In their work the authors investigated the influence of IGF1 in prostate cancer cell proliferation and migration and the role of various integrin cell adhesion molecules. Though the study is interesting per se, it was somehow unclear why the authors used two AR negative prostate cancer cell lines in their study, PC-3 and DU-145, which are not representative for the majority of prostate cancers, being AR positive and AR sensitive. Thus, confirming the most important findings in an AR positive cell line is strongly recommended in terms of finding new drug targets. In addition, since the authors concluded from their findings that there might be a cross-communication between integrin subtypes and AKT, it was not clear why the authors restricted their experiments on integrin expression to the PTEN positive DU-145 cell line. It is known that loss of PTEN as it occurs in PC-3 cells and subsequent activation of AKT frequently occurs in prostate cancer. Hence it would have been interesting to confirm the findings in PTEN negative PC-3 cells, too.

Our answer: We have now extended our studies to PC3 cells. New data are included pointing to IGF1 driven integrin expression in PC3 cells (Facs and Western blot data). Phosphorylated FAK, Rictor, Raptor and Akt are now also presented. Based on comment 3 of referee 1, experiments with the androgen-sensitive cells VCAP and LNCAP were also done. The methods section has been changed accordingly. New data sets are presented including IGF1R expression in LNCaP/VCaP (figure 1), LNCaP/VCaP cell growth in the presence of IGF1 (figure 2), integrin expression on LNCaP/VCaP (figure 5), IGF1 stimulated integrin expression on LNCaP/VCaP (figure 7) and LNCaP/VCaP cell growth blockade by integrins (figure 11).

With respect to LNCaP and VCaP cells:

The Introduction now reads (line 69): “Using human androgen-independent prostate cancer cell lines (DU-145, PC3) as well as androgen-sensitive LNCaP and VCaP cells, the present study was designed to evaluate how IGF1 is involved ….”.

Methods, Cell lines, now reads (line 76): “The human prostate tumor cell lines DU145, PC3, and LNCaP were obtained from DSMZ (Braunschweig, Germany). VCaP cell lines were from the Department of Urology and Pediatric Urology, Saarland University, Homburg/Saar, Germany. DU145, PC3, and LNCaP tumour cell lines were grown in …. (line 83) VCaP were grown in DMEM medium, supplemented with FBS, 1% penicillin/streptomycin, 2% GlutaMAX and 1% sodium pyruvate (all: Gibco/Invitrogen)”.

Results, 3.1 IGF1 activates tumor cell growth, now reads (line 224): “IGF1R expression (total and phosphorylated was verified in the androgen-sensitive cell lines LNCaP and VCaP as well (figure 1) ….. (line 234) LNCaP and VCaP did not grow well in the presence of 0% and 2% FBS, and even in the presence of 10% FBS growth activity was only moderate, compared to DU145 and PC3 cells (figure 2). Exposing LNCaP to IGF1 did not result in a significant alteration of tumor growth, whereas a significant increase in VCaP cells was seen after 72 h IGF1 incubation”.

Results, 3.2 IGF1 alters tumor cell adhesion and elevates chemotaxis, now reads (line 266): “LNCaP and VCaP did not show any chemotactic activity and were, therefore, not exposed to IGF1.

Results, 3.3 Modification of integrin surface expression, now reads (line 300): “The integrins α5 (moderately), αV, and β1 were detected on LNCaP. The subtypes α2, α3, α5, α6, αV, and β1 were detected on LNCaP cells (figure 5). …. (line 331) Alterations of the integrins α5, αV, and β1 on LNCaP and VCAP cells induced by IGF1 are shown in figure 7. The αV subtype was reduced in both cell lines after 24 h. The integrins α5 and β1 were down-regulated on VCaP as well, but not on LNCaP cells.”.

Results, 3.5 Blocking studies, now reads: “Surface expression of integrin α3, α5, αV, or β1 was blocked by their respective function associated monoclonal antibodies (line 382) and DU145, PC3, LNCaP, and VCaP growth was evaluated. Blocking integrin α3, αV, or β1 significantly suppressed tumor growth (DU145, PC3), whereas blocking α5 did not affect cell growth (shown for DU145), related to the controls (figure 11). (line 385) Blocking α5 moderately suppressed growth after 24 h in LNCaP cells. In contrast, VCaP cell growth was not affected by α5 blockage. αV blockade resulted in elevated growth activity in VCaP cells, while β1 blockade resulted in diminished growth activity in this cell line (figure 11)”.

LNCaP and VCaP data are now discussed. (line 588) “IGF1-stimulation did not alter LNCaP growth and only integrin αV was slightly reduced after 24 h incubation. We, therefore, assume that the IGF1-integrin-interaction seen in the androgen-resistant DU145 and PC3 cells is not highly relevant in the androgen-sensitive LNCaP cell line. Integrin blockage resulted in no significant difference in cell growth after 72 h. However, in contrast to LNCaP, IGF1 did activate VCaP tumor cell growth and caused diminishment of the integrins α5, αV, and β1, pointing to differences between the LNCaP and VCaP cell lines. The IGF ligand-neutralizing antibody xentuzumab has been shown to alter the proliferative activity of VCaP but not of PTEN-null LNCaP cells. The authors assumed that PTEN may be involved in IGF1 triggered cell proliferation [32]. Considering that PTEN-null PC3 cells responded well to IGF1 treatment in terms of cell growth and integrin regulation, this postulate may not hold true for our model. However, IGF1R was strongly expressed on VCaP but not on LNCaP cells, which may at least partially account for the differences observed. Since the response of the androgen-sensitive LNCaP and VCaP cell lines to IGF1-stimulation differs, androgen-sensitivity per se does not appear to be a characteristic that uniformly influences the response to IGF1-stimulation.

With respect to PC3 cells:

Results, 3.2 IGF1 alters tumor cell adhesion and elevates chemotaxis, now reads (line 261): “Horizontal migration dynamics were additionally evaluated for DU145 and PC3 cells. Whether tumor cells were pre-incubated with IGF1 for 4 (DU145 cells) or 24 h (DU145, PC3 cells), a distinct increase in motile activity was noted, compared to unstimulated controls (figure 4).

Results, 3.3 Modification of integrin surface expression, now reads (line 319): “Concerning PC3 cells, moderate differences were observed, compared to DU145 cells. Integrins α3 and α5 were diminished after 4h. However, up-regulation of α5 became evident after 24h. The integrin subtype αV considerably increased after both 4 and 24h, whereas β1 was up-regulated after 2h but down-regulated after 24h (figure 6)”.

Results, 3.4 Cell signaling pathway, now reads (line 358): “With respect to PC3 cells, IGF1 (24h stimulus) diminished integrin α3 and α5, and elevated αV. Integrin β1 was not altered (figure 10, supplement S1). The signaling proteins pRictor, pRaptor, pAkt and pFAK were all up-regulated in PC3 cells following IGF1 exposure (figure 10, supplement S1)”.

Results, 3.5 Blocking studies, now reads (line 391): “Adhesion of DU145 and PC3 to collagen or fibronectin also correlated with the integrin surface level (figure 12). Adhesion of DU145 to collagen was influenced by all blocking antibodies applied, with significant down-regulation by blocking α3, α5, or β1 and distinct elevation following αV blockade. Adhesion to fibronectin was diminished by anti-α5, -αV or -β1. The same effects were induced on PC3 adhesion to fibronectin. However, adhesion of PC3 to collagen was down-regulated by all anti-α3, -α5, or -αV antibodies (figure 12)”.

We have also included in ”Discussion” (line 494): Slight differences in integrin modulation have also been noted between DU145 and PC3 cells. Particularly, αV was enhanced on PC3 cells after 4h and 24h IGF1 incubation. Hypothetically, this difference might explain why blocking αV surface expression correlated with an increased adhesion of DU145 cells but decreased adhesion of PC3 to immobilized collagen…… (line 509) Since differences are seen in initial integrin expression levels, with β3 verified on DU145 but not on PC3 cells, integrin trafficking, if it does take place, must also differ. We have shown that integrin protein expression in response to IGF1 does differ between DU145 and PC3 cells. Differences in the genetic pattern of different tumor types may possibly be involved in modulating particular integrin subtypes since DU145 harbors mutations in CDKN2A, RB1, and TP53, whereas PC3 harbors mutations in PTEN and TP53. ”.

Comment 2: Cell growth assays were performed in medium with 0%, 2% and 10 % FBS. The authors speculated that FBS influences or masks the effect of IGF1, which is most probably the case. Have the authors measured how much IGF1 is in the FBS? It would have been interesting to see the effect of neutralizing antibodies to integrins on cell growth in 0%, 2% and 10% FBS.

Our answer: In regard to this comment we have included further data. Results, 3.2 IGF1 alters tumor cell adhesion and elevates chemotaxis, now reads (line 264). “The FBS-content (2 versus 10%) did not influence the effect of IGF1 on horizontal migration of DU145 or PC3 cells (figure 4). LNCaP and VCaP cells did not show any chemotactic activity and were, therefore, not exposed to IGF1.

Results, 3.5 Blocking studies, now reads (line 421): “The effects of integrin blockade on tumor growth did not depend on IGF1- or FBS-concentration. Figure 14A depicts DU145 cell growth data derived from cultures with 2 versus 10% FBS and with or without IGF1-activation. The IGF1-concentration did not influence integrin dependent alterations of DU145 chemotaxis, as shown in figure 14B. Adhesion of DU145 cells to collagen and fibronectin in the presence of β1 function blocking antibodies was similarly blocked, whether tumor cells were grown in 2 or 10% FBS (figure 14C)”.

Comment 3: With regard to the experiments with neutralizing antibodies it was not clear if they were done in the absence or presence of IGF-1. To my mind, the neutralizing antibodies should have been used also in the adhesion experiments to give an information on the role of the different integrins in adhesion to collagen and fibronectin in prostate cancer cells. As far as I understood these experiments were conducted in the absence of IGF1. To my mind, however, it would be rather interesting to see the effects in the absence and presence of IGF1.

Our answer: Adhesion of both DU145 and PC3 cells to fibronectin and collagen in the presence of integrin moAbs was evaluated. The data are presented in figure 12. We also investigated whether IGF1 may influence chemotaxis and whether the FBS-content may influence adhesion. The respective data are shown in figures 14B and C. Please note that the chemotaxis could only be conducted in the presence of 2% serum in the upper chamber, which was necessary to establish a serum gradient.

Comment 4: To my mind, Figure 4 could be omitted since the scratch wound assay does not really give any additional valuable information and – besides – has only been done in DU-145 cells.

Our answer: The scratch wound assay was also done with PC3 cells with the same response to IGF1 as seen with DU145 cells. The data have now been included. Results, 3.2 IGF1 alters tumor cell adhesion and elevates chemotaxis, now reads (line 261): “Horizontal migration dynamics were additionally evaluated for DU145 and PC3 cells. Whether tumor cells were pre-incubated with IGF1 for 4 (DU145 cells) or 24 h (DU145, PC3 cells), a distinct increase in motile activity was noted, compared to unstimulated controls (figure 4)”.

Comment 5: Flow cytometry and Western blotting revealed different results with regard to the expression of integrin subtypes. Which conditions were used for cell lysis in Western blot experiments? Do the lysates really only contain intracellular integrins or rather a mixture of intra- and extracellular integrins? Could the discrepant results be a consequence of using different antibody clones for Western blotting and flow cytometry? In the discussion, the authors claimed that IGF1 might cause a translocation of a3 and aV from the cell membrane into the cytoplasm and a translocation of a5 from inside the cell to the surface membrane. Would it be possible to show this translocation by immunofluorescence?

Our answer: The referee is correct. Protein data are related to membranous and intracellular integrins. To make this point clear and prevent confusion, Results, 3.3 Modification of the integrin expression, last paragraph, starts with (line 341) “Total integrin protein content ….”.

Since there was no homogenous integrin profiling following IGF1 exposure (Western blot versus Facs analysis), we speculated on integrin translocation processes. In fact, integrin trafficking is a well-known phenomenon. We do not believe that the differences are caused by the use of different antibodies, since all studies were accompanied by controls. Nevertheless, we agree with the referee’s comment that ongoing experiments are necessary to verify our assumption. We also agree that confocal laser scanning microscopy might be the method of choice. Unfortunately, this method has not yet been established in our lab. Meanwhile, the manuscript has been extensively widened and now includes 15 figures. Investigating integrin translocation processes would therefore not be possible in a manageable time frame. However, to deal with the referee’s comment, we have included in the discussion section (line 509): “Since differences are seen in initial integrin expression levels, with β3 verified on DU145 but not on PC3 cells, integrin trafficking, if it does take place, must also differ. We have shown that integrin protein expression in response to IGF1 does differ between DU145 and PC3 cells. Differences in the genetic pattern of different tumor types may possibly be involved in modulating particular integrin subtypes since DU145 harbors mutations in CDKN2A, RB1, and TP53, whereas PC3 harbors mutations in PTEN and TP53.

We have rewritten the Simple Summary and the Abstract to conform with Cancer’s length requirements and to incorporate added experimental work suggested by the reviewers.